# Functions of Breast Cancer Predisposition Genes: Implications for Clinical Management

**DOI:** 10.3390/ijms23137481

**Published:** 2022-07-05

**Authors:** Akiyo Yoshimura, Issei Imoto, Hiroji Iwata

**Affiliations:** 1Department of Breast Oncology, Aichi Cancer Center Hospital, 1-1 Kanokoden, Chikusa-ku, Nagoya 464-8681, Japan; hiwata@aichi-cc.jp; 2Aichi Cancer Center Research Institute, 1-1 Kanokoden, Chikusa-ku, Nagoya 464-8681, Japan; iimoto@aichi-cc.jp

**Keywords:** hereditary breast cancer, cancer predisposition gene, function, BRCA, cancer prevention

## Abstract

Approximately 5–10% of all breast cancer (BC) cases are caused by germline pathogenic variants (GPVs) in various cancer predisposition genes (CPGs). The most common contributors to hereditary BC are *BRCA1* and *BRCA2*, which are associated with hereditary breast and ovarian cancer (HBOC). *ATM*, *BARD1*, *CHEK2*, *PALB2*, *RAD51C*, and *RAD51D* have also been recognized as CPGs with a high to moderate risk of BC. Primary and secondary cancer prevention strategies have been established for HBOC patients; however, optimal preventive strategies for most hereditary BCs have not yet been established. Most BC-associated CPGs participate in DNA damage repair pathways and cell cycle checkpoint mechanisms, and function jointly in such cascades; therefore, a fundamental understanding of the disease drivers in such cascades can facilitate the accurate estimation of the genetic risk of developing BC and the selection of appropriate preventive and therapeutic strategies to manage hereditary BCs. Herein, we review the functions of key BC-associated CPGs and strategies for the clinical management in individuals harboring the GPVs of such genes.

## 1. Introduction

Breast cancer (BC) is the most common cancer affecting women globally [1,2,3] and the risk factors for BC include acquired environmental and genetic factors [4,5,6,7]. The main environmental factors include aging, obesity, alcohol consumption, smoking, and an increased exposure to estrogens, such as early menarche, late menopause, late delivery, and postmenopausal hormone replacement therapy combined with estrogen plus progestin. Up to 5–10% of all BC cases are considered to be caused by germline pathogenic variants (GPVs) in various cancer predisposition genes (CPGs) (Figure 1). Such hereditary BCs exhibit an autosomal dominant inheritance [8,9,10].

The most common cause of hereditary BC is a GPV in *BRCA1* or *BRCA2* (*BRCA1/2*) associated with hereditary breast and ovarian cancer (HBOC). *BRCA1* is also known to be inactivated in sporadic breast cancer by somatic promoter hypermethylation [11], suggesting that both the germline and somatic inactivation of these genes play a critical role in BC tumorigenesis.

BC is one of the most preventable types of cancer and the primary prevention strategies for BC include eliminating the causes of the disease through education and public health programs, risk-reducing surgery, and chemoprevention. Secondary prevention aims to reduce cancer mortality by early detection using appropriate screening methods. Considering the high risks of developing BCs, individuals harboring GPVs in the BC-associated CPGs are excellent candidates for risk minimization via primary prevention or the enhanced detection of certain cancers at their earliest and most treatable stages.

Several BC-associated CPGs have been identified. Recently, the results of two large-scale case-control studies analyzing the association between GPVs in BC-associated CPGs and BC risk were reported. Dorling et al. analyzed a panel of 34 known or suspected BC-associated CPGs in 60,466 women with BC and 53,461 controls [12], whereas Hu et al. analyzed a panel of 28 known or suspected BC-associated CPGs in 32,247 women with BC and 32,544 controls [13]. Eight genes, *ATM*, *BRCA1*, *BRCA2*, *BARD1*, *CHEK2*, *PALB2*, *RAD51C*, and *RAD51D*, were observed to be significantly associated with a BC risk in at least one of the two studies [12,13] (Table 1). *BRCA1* and *BRCA2* are high-risk CPGs for BC. *ATM*, *BARD1*, *CHEK2*, *PALB2*, *RAD51C*, and *RAD51D* have also been recognized as CPGs with high to moderate risks for BC. Figure 2 shows the total prevalence of 12 genes (three high and five moderate penetrance genes and four syndromic genes) in the BC cases reported by Dorling et al.

Primary and secondary prevention strategies have been established for HBOC; however, optimal preventive strategies for most other hereditary BCs have not yet been established and are sometimes considered based on individual family history [14] (Table 2). Most BC-associated CPGs are involved in DNA damage repair pathways and cell cycle checkpoint mechanisms, and function together in these physiological cascades [9,15,16]; therefore, a fundamental understanding of the disease drivers in the cascades would facilitate the accurate estimation of the genetic risk of BC development and the selection of appropriate preventive and therapeutic strategies for managing hereditary BCs.

This review outlines the functions of eight CPGs with high to moderate risks of BC, which collaborate in DNA damage repair pathways and/or cell cycle checkpoints, and the clinical management of individuals harboring GPVs in the genes. The *CDH1*, *PTEN*, *STK11*, and *TP53* syndromic genes, which cause hereditary diffuse gastric cancer (HDGC) [17], Cowden disease/PTEN hamartoma tumor syndrome (PHTS) [18], Peutz–Jeghers syndrome (PJS) [19], and Li–Fraumeni syndrome (LFS) [20], respectively, and which are also BC-associated CPGs, were excluded from this review.

## 2. High-Penetrance Genes in Breast Cancer

*BRCA1*, *BRCA2,* and *PALB2* are the most widely known high-penetrance genes involved in severe BC risk. *BRCA1*- and *BRCA2*-associated HBOC accounts for approximately 50% of hereditary BC [8] (Figure 2).

### 2.1. BRCA1 and BRCA2

#### 2.1.1. Functions

The first evidence of the existence of a BC susceptibility gene encoding a DNA repair enzyme on chromosome 17 was provided by linkage studies in 1990 [21], and *BRCA1* (BReast-CAncer susceptibility gene 1) on chromosome 17 was identified in 1994 [22]. Following the identification of *BRCA1*, *BRCA2* (BReast-CAncer susceptibility gene 2), located on chromosome 13, was identified in 1995 [23] (Figure 1). 

*BRCA1* and *BRCA2* play crucial roles in DNA damage repair, specifically in the repair of DNA double-strand breaks (DSBs), which contributes to the maintenance of chromosome structure through homologous recombination (HR) repair, cell cycle checkpoint activation, and DNA replication fork protection [24,25,26].

HR Repair

There are two major mechanisms of the repair of DSBs, which represent one of the most serious types of DNA damage and must be repaired to preserve chromosomal integrity: non-homologous end-joining (NHEJ) and HR. As NHEJ directly ligates broken ends without requiring extended homologies, DNA sequence errors are more likely to occur, and the repaired products often harbor sequence alterations. This repair pathway is active throughout the cell cycle, particularly in the G0/G1 phase when sister chromatids are absent. Contrastingly, HR is the main pathway used strictly in late S or G2 phase cells and can accurately repair the damage using intact homologous sister chromatids as a sequence homologous template for the DNA repair reaction [9,27] (Figure 3a).

After DSBs occur, the HR is sequentially organized. In mammals, the free DNA ends produced by DSBs are initially recognized by the MRE11-RAD50-NBS1 (MRN) protein complex. The MRN complex recruits and activates ATM, which phosphorylates several crucial proteins mediating the DNA damage response (DDR). Active ATM monomers phosphorylate H2AX in regions of the DSBs and create a platform to recruit BRCA1 [28]. In the second step of HR, the so-called DSB end resection, a CtBP-interacting protein, in conjunction with the MRN complex, catalyzes the 5′-3′ end resection at DSBs to generate a long 3′ single-stranded DNA (ssDNA) tail [28]. Further resection is completed by exonuclease 1 and DNA2 nuclease/helicase, in conjunction with BLM helicase [28]. The resulting ssDNA tail is protected from degradation by the replication protein A (RPA). Meanwhile, RPA-covered ssDNA recruits ATR–ATRIP to activate the DNA damage checkpoint, leading to cell cycle arrest in the G2/M phase. Activated ATR phosphorylates PALB2, which, in turn, favors the PALB2-BRCA1 interaction [29]. Phosphorylated BRCA1 is concentrated in the focal areas of DNA damage, and the BRCA1-BARD1 heterodimer interacts with PALB2 and BRCA2 to recruit RAD51, an essential mediator in the HR repair pathway, resulting in the BRCA2-mediated displacement of RPA with RAD51, homology search, strand invasion toward a homologous template, and the formation of a D-loop for HR [9,15,30]. DNA polymerase δ (Pol δ) elongates the invaded strand by using homologous DNA as a template. Once repair synthesis occurs, the D-loop is resolved through synthesis-dependent strand annealing (SDSA) or Holliday junction-containing intermediates.

Carriers of GPVs in *BRCA1/2* usually inherit a single variant copy in the germline. The “two-hit” paradigm of tumor suppression advocated by Knudson is widely supported and considered to lose its function when the second copy is inactivated by somatic variants or epigenetic events [31]. A loss of heterozygosity (LOH) of the wild-type allele in the carriers of *BRCA1/2* GPVs leads to a loss of function (LOF) of these genes. Dysfunction of *BRCA1/2* leads to cancer susceptibility via the induction of chromosomal instability and mutagenesis [32,33]. Several studies have shown that environmental and endogenous toxins, such as aldehydes, induce genomic instability, and *BRCA1* and/or *BRCA2* protect against such toxicity and suppress tumorigenesis [34,35,36]. In addition, endogenous acetaldehyde toxicity may induce BRCA2 haploinsufficiency [35]. According to such reports, aldehyde accumulation may modulate tissue-specific cancer progression in *BRCA1/2* GPV carriers, with implications for public health [34] 

DNA Damage-induced Cell Cycle Checkpoints

Cell cycle checkpoints represent an important regulatory mechanism for cell survival, wherein the cell cycle process is inhibited in the presence of unresolved DNA damage; the cell cycle is inhibited until the damage is fully repaired [9,26] (Figure 3b). In response to DNA damage, cell cycle checkpoints can be activated in the G1, S, and G2/M phases, and the processes that detect and signal DNA damage to downstream effectors depend on the phases of the cell cycle. *BRCA1* contributes to the activation of DNA damage checkpoints in the G1/S, intra-S, and G2/M phases, and various *BRCA1* complexes regulate different phases of the cell cycle. Consequently, the LOF of *BRCA1* causes checkpoint abnormalities [30]; however, the role of *BRCA2* in the regulation of cell cycle checkpoints remains unclear. Notably, *BRCA1* is essential for both the initial activation and G2/M checkpoint maintenance, whereas *BRCA2* and *PALB2* appear to be more important for its maintenance during DNA damage [37].

R-loop Processing and Transcription

During transcription, nascent RNA binds strongly to the template DNA strand, forming a unique RNA–DNA hybrid structure that displaces the non-template ssDNA [28]. This three-stranded nucleic acid transition is known as the R loop. Unscheduled R-loop formation can lead to genomic instability in a variety of ways, such as ssDNA formation, the induction of transcription block/slow down (transcription stress), replication stress, and DSBs.

Multiple proteins involved in transcription recognize the R-loop structure and subsequently resolve it, resulting in transcription-associated HR repair (TA-HRR) [38] (Figure 3c). *BRCA1* and *BRCA2* play roles in R-loop turnover, working with several R-loop processing factors at the promoter-proximal pausing (PPP) sites or transcription termination (TT) sequences of expressed genes. In *BRCA1/2*-deficient cells, R-loop accumulation is a major cause of replication stress and DNA damage, resulting in chromosomal fragility. Consequently, the accumulation of R-loop associated with *BRCA1/2* inactivation has been reported to be associated with BC development [39,40]. It has become clear that BRCA1/2 dysfunction causes genomic instability, leading to carcinogenesis; however, why BC and OC develop at high rates is still not fully understood. Several plausible theories have been proposed regarding the relationship between estrogen and organ-specific mechanisms of carcinogenesis in *BRCA1/2* GPV carriers. Differentiated luminal cells in the normal mammary gland express estrogen receptor (ER) and progesterone receptor (PR), whereas the ovarian tissue is ER-negative [41]. Notably, ER and PR are reportedly expressed in fallopian tube epithelial cells, which are considered the origin of epithelial OC [42].

DNA Replication Fork Protection

DNA replication is the process of producing two identical DNA replicas from an initially double-stranded DNA molecule during cell division. Proper control of DNA replication during each cell cycle stage is essential for genomic stability, and during DNA replication, a replication fork is formed by unraveling the double-helical DNA strands. 

Replication is frequently arrested by various genotoxic stressors, including DNA damage. Stalled forks are unstable structures that cause genomic instability, which is a hallmark of tumorigenesis, if not properly processed; therefore, stalled forks are protected by various mechanisms, including replication fork reversal, until the reactivation of the paused forks. The reversed replication forks are protected from MRE11 nuclease degradation by *BRCA1*, *BRCA2*, *RAD51*, and components of the Fanconi anemia (FA) complex, such as FANCA, FANCD2, and FANCJ (BRIP1) [43,44,45] (Figure 3d). Failure to protect and reactivate the replication fork exposed to replication stress may induce genomic instability and promote carcinogenesis [45]. 

Removal of Estrogen-induced Pathological Topoisomerase II–DNA Complexes to Ensure Genome Integrity

The binding of estrogens to the ER transiently induces DSBs via topoisomerase II (TOP2), followed by the enzymatic religation of the broken strands to untangle the DNA in transcriptional regulatory sequences [46], and also controls gene transcription (Figure 4).

In the process, TOP2 forms TOP2–DNA cleavage complex intermediates (TOP2ccs) but it frequently fails to complete the religation step, resulting in the formation of pathologically stable TOP2ccs as 5′ adducts. BRCA1 repairs pathological TOP2ccs by promoting the estrogen-induced recruitment of MRE11 to the TOP2cc sites.

Similarly, TOP2β-mediated DSBs occur during AR transcription in the prostate, suggesting that BRCA1 dysfunction in ER- and AR-expressing organs contributes to the DNA repair defects that result in ER- and AR-associated tumorigenesis [47].

R-loop Regulation of the Enhancer Region of ESR1 by BRCA1

The mechanism of ESR1 transcriptional dysregulation due to BRCA1 dysfunction, resulting in basal-like BC has been previously reported [48] (Figure 5).

During normal mammary development, ER-negative and PR-negative luminal progenitors differentiate into ER-positive mature luminal cells [49]. Luminal fate genes, such as *ESR1*, which encodes the estrogen receptor (Erα), are involved in the differentiation of luminal progenitors into mature luminal cells. *BRCA1*-associated BCs have a high frequency of basal-like and triple-negative BC (TNBC) derived from luminal progenitor cells. BRCA1 mutation causes an elevated R-loop at a putative transcriptional enhancer upstream of the *ESR1* gene and reduces the transcription of the corresponding neighboring genes, such as *ESR1*, *CCDC170*, and *RMND1*. The resulting abnormal expansion of ER-negative luminal progenitors may be a target for other oncogenic hits, leading to basal-like *BRCA1*-associated BC formation [39,50].

#### 2.1.2. Prevalence and Risk of Developing Cancers

In two large-scale case-control studies reported by Dorling et al. and Hu et al., GPVs in *BRCA1* were detected in 1.05% and 0.85% of patients with BC and 0.11% and 0.11% of the controls, respectively, whereas GPVs in *BRCA2* were detected in 1.54% and 1.29% of patients with BC and 0.26% and 0.24% of the controls, respectively [12,13] (Table 1).

The odds ratios of *BRCA1* and *BRCA2* for BC risk were reported to be 10.57 and 5.85, respectively, by Dorling et al., and 7.62 and 5.23, respectively, by Hu et al. In addition, the odds ratios for female *BRCA1* GPV carriers were higher for TNBC (56.80 and 42.88, respectively) than for ER-positive BC (3.92 and 3.39, respectively) (Table 1).

The cumulative BC risk by 80 years was reported to be 72% and 69% for female *BRCA1* and *BRCA2* GPV carriers, respectively [51]. According to two recent studies, *BRCA1/2* GPVs yielded a lifetime risk of approximately 50% at 80 years [12,13], which was lower than that reported in previous family-based studies. Male *BRCA1* and *BRCA2* GPV carriers are also at high risk for the development of BC, with risk estimates of ~1–5% and 5–10%, respectively, compared with the general male population, with a lifetime risk of ~0.1% [1].

It is estimated that 10–15% of patients with OC harbor GPVs in *BRCA1/2* [52,53], and the cumulative lifetime risks are 44% and 17% for *BRCA1* and *BRCA2* GPV carriers, respectively [51].

In addition, *BRCA1/2* GPV carriers have increased risks of developing prostate cancer, pancreatic cancer, and malignant melanoma. In a meta-analysis, the pooled relative risks (RRs) for prostate cancer were 4.35 and 1.18 for non-Ashkenazi European ancestry *BRCA2* and *BRCA1* GPV carriers, respectively [54]. The risks of developing prostate cancer by the age of 75 years were reportedly 21% and 27% for *BRCA1* and *BRCA2* GPV carriers, respectively, and the risks of developing pancreatic cancer by 75 years were reportedly 1–3% and 3–5% for *BRCA1* and *BRCA2* GPV carriers, respectively [55,56,57]. In particular, the RR of pancreatic cancer is high when there are one or more first-degree relatives with pancreatic cancer [58].

Biallelic variants in the genes involved in the FA/BRCA DNA repair pathway cause FA. FA is a genomic instability syndrome characterized by a predisposition to congenital abnormalities, early-onset bone marrow failure, and cancer. It is often an autosomal recessive genetic disorder [59].

#### 2.1.3. Medical Management for Cancer Prevention

For BC surveillance in women with *BRCA1/2* GPVs, the latest version of the National Comprehensive Cancer Network (NCCN) Clinical Practice Guidelines for Genetic/Familial High-Risk Assessment: Breast, Ovarian, and Pancreatic, Version 2.2022 (https://www.nccn.org/home, Last accessed on 16 April 2022) recommends BC awareness from an age of 18 years and clinical breast examination every 6–12 months from an age of 25 years. Between the ages of 25–29 years, annual breast magnetic resonance imaging (MRI) with contrast is recommended. Between the ages of 30–75 years, an annual mammogram with consideration of tomosynthesis and breast MRI with contrast is recommended. The management should be considered on an individual basis for *BRCA1/2* GPV carriers aged > 75 years (Table 2).

Risk-Reducing Mastectomy (RRM)

Based on the results of meta-analyses, it is certain that contralateral RRM (CRRM) in BC patients with *BRCA1/2* GPVs reduces the risk of developing BC in the contralateral breast [60,61,62]; however, the survival benefit of CRRM remains uncertain because the effect of risk-reducing salpingo-oophorectomy (RRSO) cannot be excluded [60,61,62].

Furthermore, it is certain that bilateral RRM (BRRM) in *BRCA1/2* GPV carriers without BC reduces the risk of developing BC in bilateral breasts; however, according to the results of some studies, BRRM is not significantly associated with improved survival [60,63,64].

Therefore, it is important to consider individual values and shared decision-making in the RRM options [14].

Risk-Reducing Salpingo-Oophorectomy (RRSO)

*BRCA1/2* GPV carriers are at high risk of developing OC. In contrast to those of BC, the outcomes of advanced OC are poor, and there are considerable limitations to the effective early detection of OC. 

Previous prospective cohort studies and meta-analyses have shown that RRSO is effective in preventing OC development and prolonging the overall survival (OS) of *BRCA1/2* GPV carriers [65].

According to the NCCN guidelines, RRSO is usually advised for *BRCA1* GPV carriers between the ages of 35 and 40 years, post-childbearing. As *BRCA2* GPV carriers develop OC on an average 8–10 years later than *BRCA1* GPV carriers, it is reasonable to postpone RRSO until the age of 40–45 years for *BRCA2* GPV carriers unless family members are diagnosed at an earlier age [14].

Currently, primary prevention is limited to risk-reducing surgeries for unaffected organs; therefore, effective non-surgical chemopreventive strategies need to be established.

Tamoxifen

In a meta-analysis, tamoxifen was found to significantly reduce the incidence of CBC among *BRCA1/2* GPV carriers with primary unilateral BC (RR, 0.56) [64]. Similarly, preventive effects of tamoxifen have been observed in *BRCA1* GPV (RR, 0.47) and *BRCA2* GPV (RR, 0.39) carriers [66].

However, previous data are not consistent regarding the optimal dose and duration of treatment. In addition, its effect on OS in female *BRCA1/2* GPV carriers, particularly in women who have also undergone oophorectomy, remains unclear.

Denosumab: Anti-RANK-L Monoclonal Antibody

*BRCA1* GPV carriers have a high incidence of basal-like BC and basal-like BC is derived from ER-negative and PR-negative luminal progenitor cells. It has been reported that these luminal progenitor cells are stimulated by the paracrine action of RANK-L secreted by the surrounding mature luminal cells, and this proliferative signal leads to the development of BC. Denosumab, a RANK-L inhibitor, has the potential to prevent *BRCA1*-associated BC, and clinical trials on denosumab are ongoing [67,68] (Figure 5).

#### 2.1.4. Treatment

Anthracycline-based Chemotherapy/Taxanes

Conventionally, anthracycline- and taxane-based (AC-T) regimens are used for BC treatment. Anthracyclines induce DSBs by inhibiting TOP2. In vitro data suggest that cells lacking the BRCA1 or BRCA2 functions are sensitive to agents such as anthracyclines, which cause DSBs and consequently increase apoptosis [69,70,71].

In contrast, taxanes are anti-microtubule agents that exert their effects by inhibiting mitotic spindle depolymerization and tubulin polymerization [71]. Preclinical data show that the inhibition of BRCA1 leads to a resistance to anti-microtubule agents [71,72,73].

Neoadjuvant studies have investigated the pathological complete response (pCR) rate after AC-T in *BRCA1/2* GPV carriers and non-carriers. The *BRCA1* GPV-positive status and ER-negative status in patients with BC are reportedly independently associated with an increased pCR rate [71,74].

Platinum-based Anticancer Agents

*BRCA*-associated BCs are more sensitive to platinum-based chemotherapy [22,23]. The cytotoxicity of platinum-based drugs involves the binding of platinum to DNA and forming cross-links, which interferes with DNA replication and transcription, resulting in DSBs. Accordingly, *BRCA1/2*-deficient BC is expected to be particularly sensitive to platinum-based agents more than *BRCA1/2* wild-type BC [71,75,76].

A retrospective study using the Poland registry in 2010 reported, for the first time, the sensitivity of patients with *BRCA1* GPV to a neoadjuvant platinum agent [77]. Among 102 patients with *BRCA1* GPV, a higher rate of pCR (83%) was observed after treatment with cisplatin (75 mg/m^2^ every three weeks for four cycles) compared to that (22%) for AC (doxorubicin and cyclophosphamide) or FAC (fluorouracil, doxorubicin, and cyclophosphamide); however, a randomized phase II study of neoadjuvant cisplatin versus AC in 118 HER2-negative BC patients with *BRCA1/2* GPV (TBCRC 031) in 2020 demonstrated that the pCR or residual cancer burden (RCB) 0/1 was not significantly higher in the cisplatin-treated group than in the AC-treated group [78]. The pCR rate was 18% with the cisplatin and 26% with the AC (RR, 0.70; 90% confidence interval [CI], 0.39 to 1.2), and the RCB 0/1 was 33% with the cisplatin and 46% with the AC (RR, 0.73; 90% CI, 0.50 to 1.1). A meta-analysis showed that the addition of platinum to chemotherapy regimens in the neoadjuvant setting increases the pCR rate in *BRCA1/2*-associated TNBC patients (58.4%, 93/159 cases) as compared to wild-type TNBC patients (50.7%, 410/808 cases), although the difference was not significant [79].

The TNT and CBCSG006 trials examining the benefit of the first-line platinum agents in patients with metastatic TNBC have included those with *BRCA1/2* GPV [71,80,81]; however, no randomized controlled trials have investigated the efficacy of platinum-based chemotherapy in patients with *BRCA1/2*-related advanced BC.

The TNT trial compared first-line carboplatin with docetaxel [73]. In the subgroup analysis of the TNBC patients with *BRCA1/2* GPVs, carboplatin showed twice the overall response rate (ORR) of docetaxel (68% vs. 33%, *p* = 0.03). Progression-free survival (PFS) also favored the carboplatin (6.8 months vs. 4.4 months), but no difference was found in OS, probably because of the crossover design of the trial. 

In the CBCSG006 trial, the cisplatin plus gemcitabine (GP) regimen showed an efficacy superior to that of the paclitaxel plus gemcitabine (GT) regimen (hazard ratio = 0.692) as the first-line treatment for metastatic TNBC [74]. In an additional biomarker assessment, patients with *BRCA1/2* GPVs had a numerically higher ORR (83.3% vs. 37.5%, *p* = 0.086) and prolonged PFS (8.90 vs. 3.20 months, *p* = 0.459) in the GP arm than in the GT arm.

CDK4/6 Inhibitors

A study presented at the 2021 San Antonio Breast Cancer Symposium (SABCS) showed that *BRCA2* GPV carriers had poor outcomes following treatment with first-line CDK4/6 inhibitors plus endocrine therapy in 4640 patients with BC whose samples were subjected to germline and matched tumor tissue sequencing using MSK-IMPACT from April 2014 to May 2021. As *RB1* and *BRCA2* are located near the same chromosome, an *RB1* loss associated with the LOH of *BRCA2* is considered one of the mechanisms underlying CDK4/6 resistance [82]. 

In retrospective real-world data with a total of 217 individuals including 15 patients with *BRCA1/2* (*n* = 10), *ATM* (*n* = 4), and *CHEK2* (*n* = 1) GPVs, *BRCA1/2-ATM-CHEK2* GPVs were suggested to be associated with poor outcomes in advanced BC treated with CDK4/6 inhibitors; however, there were no randomized controlled trials for CDK4/6 inhibitors in patients with GPVs in the HR genes [83].

PARP Inhibitors

PARP inhibitors exhibit anticancer activities based on the theory of “synthetic lethality” between PARP inhibition and the LOF of BRCA1/2 due to pathogenic variants or depletion. Several PARP inhibitors with various PARP trapping activities have been developed. Here, we summarize the clinical studies of PARP inhibitors, including olaparib, talazoparib, veliparib, niraparib, and rucaparib, in *BRCA1/2*-associated BC patients, in Table 3, and briefly mention several representative phase III trials among them.

Olaparib is a PARP-1, -2, and -3 inhibitor. HER2-negative metastatic BC (MBC) and early BC with *BRCA1/2* GPVs have demonstrated a high sensitivity to Olaparib, based on the results of the OlympiAD and OlympiA trials, respectively [84,85]. 

In the OlympiAD trial, an olaparib monotherapy was clinically superior to the standard chemotherapy of physician’s choice (TPC) in terms of both PFS (7.0 vs. 4.2 months) and ORR (59.9% vs. 28.8%). There was no statistically significant improvement in the OS [84]. 

In the OlympiA trial for patients with high-risk BC after the local treatment and adjuvant chemotherapy, additional adjuvant olaparib for 12 months provided a significantly longer invasive or distant disease-free survival (DFS) than the placebo. The three-year invasive DFS was 85.9% in the olaparib group and 77.1% in the placebo group [85].

Talazoparib is a PARP-1 and -2 inhibitor with powerful PARP trapping. An EMBRACA trial comparing talazoparib and TPC in MBC patients with *BRCA1/2* GPVs showed that the PFS and ORR were significantly better in the talazoparib group than in the TPC group (8.6 vs. 5.6 months, 62.6% vs. 27.2%, respectively) [86].

Veriparib is a PARP-1 and -2 inhibitor with the weakest PARP trapping activity, which has been essentially developed in combination with platinum-based chemotherapy. In the BrighTNess trial, the effects of the addition of carboplatin with and without veriparib to the standard neoadjuvant combination of paclitaxel followed by AC were evaluated in 634 TNBC patients including 92 patients with *BRCA1/2* GPVs [87]. The stratified results for the pCR rate for patients with *BRCA1/2* GPVs treated with paclitaxel alone, those treated with paclitaxel plus carboplatin, and those treated with paclitaxel plus carboplatin plus veliparib were 41%, 50%, and 57%, respectively.

**Table 3 ijms-23-07481-t003:** Clinical studies of PARP inhibitors in *BRCA1/2*-associated breast cancer.

Trial	Type of Study	Patients	Arms	Results
**Olaparib**				
**Neoadjuvant setting**				
GeparOLA [88]	Ph. II	102 HER2 negative-BC pts with HRD tumors	Paclitaxel and olaparib followed by EC (*n* = 65) Paclitaxel and carboplatin followed by EC (*n* = 37)	Olaparib arm pCR = 55.1% Carboplatin arm pCR = 48.6%
**Adjuvant setting**				
OlympiA [85]	Ph. III RCT	1836 pts with *BRCA1/2* GPV post (neo)adjuvant chemotherapy	Olaparib 300 mg Placebo for 12 mo	Olaparib IDFS = 85.9% Placebo IDFS = 77.1%
**Advanced or Metastatic Setting**				
Tutt A, et al. [89]	Ph. II non-randomised	54 MBC pts with *BRCA1/2* GPV	Olaparib 400 mg Olaparib 100 mg	ORR = 41% ORR = 22%
Kaufman B, et al. [90]	Ph. II Single arm	298 solid tumor pts (62 BC) with *BRCA1/2* GPV >3 lines of chemotherapy for MBC	Olaparib 400 mg	RR = 12.9% (8/62 pts)
OlympiAD [84]	Ph. III RCT	302 MBC pts with *BRCA1/2* GPV <2 lines of chemotherapy for MBC	Olaparib 300 mg TPC	Olaparib ORR = 59.9% mPFS = 7.0 mo TPC ORR = 28.8% mPFS = 4.2 mo
**Niraparib**				
**Advanced or Metastatic Setting**				
Sandhu SK, et al. [91]	Ph. I dose-escalation	100 solid tumors including 22 MBC	Niraparib 30–400 mg daily	Maximum-tolerated dose is 300 mg daily
**Rucaparib**				
**Advanced or Metastatic Setting**				
Drew Y, et al. [92]	Ph. II dose escalation IV → oral study	78 solid tumors pts including 23 pts with *BRCA1/2* GPV	Rucaparib IV 4–18 mg →oral 92–600 mg twice daily	Well-tolerated doses as oral 480 mg daily.
Wilson RH, et al. [93]	Ph. I dose-escalation in combination with chemotherapy	85 pts with advanced solid tumors including 7 pts with *BRCA1/2* GPV	Rucaparib IV12–24 mg →oral 80–360 mg + chemotherapy	Maximum-tolerated dose for the combination was oral 240 mg daily rucaparib and carboplatin
Miller K, et al. [94]	Ph. II RCT	128 pts with TNBC or *BRCA*-associated BC (*n* = 22) with residual tumor post neoadjuvant chemotherapy	Cisplatin 75 mg/m^2^ ± Rucaparib 25–30 mg IV days 1 to 3 (4 cycles) → oral rucaparib 100 mg weekly	Cisplatin alone 2-yr DFS = 54.2% Cisplatin + rucaparib 2-yr DFS = 64.1%
**Talazoparib i**				
**Neoadjuvant setting**				
Litton JK, et al. [95]	Ph. II	20 HER2 negative BC pts with *BRCA1/2* GPV	Talazoparib 1 mg for 6 mo	RCB 0 (pCR) = 53% RCB 0/1 = 63%
**Advanced or Metastatic Setting**				
EMBRACA [86]	Ph. III RCT	431 advanced/metastatic BC pts with *BRCA1/2* GPV <3 lines of chemotherapy for MBC	Talazoparib 1 mg TPC	Talazoparib ORR = 62.6% mPFS = 8.6 mo TPC ORR = 27.2% mPFS = 5.6 mo
**Veriparib**				
**Neoadjuvant setting**				
I SPY2 [96]	Ph. II adaptive randomized trial	Stage II or III TNBC (*n* = 116) veriparib group (*n* = 72) including 12 pts with *BRCA1/2* GPV control group (*n* = 44) including 3 pts with *BRCA1/2* GPV	Carboplatin/paclitaxel + placebo (CP) Carboplatin/paclitaxel + veriparib 50 mg (VCP) All patients received followed by AC	CP pCR = 26% VCP pCR = 51%
BrighTNess [87]	Ph. III RCT	Stage II or III TNBC (*n* = 634) including 92 pts with *BRCA1/2* GPV	Paclitaxel CP VCP All patients received followed by AC	Paclitaxel pCR = 31% CP pCR = 58% VCP pCR = 53%
**Advanced or Metastatic Setting**				
BROCADE [97]	Ph. II RCT	290 advanced/metastatic BC with *BRCA1/2* GPV	CP VCP Veliparib 120 mg + temozolomide (VT)	CP ORR = 61.3% mPFS = 12.3 mo VCP ORR = 78% mPFS = 14.1 mo VT ORR = 28.6% mPFS = 7.4 mo

BC, breast cancer; CP, carboplatin/paclitaxel; DFS, disease-free survival; EC, epirubicin and cyclophosphamide; GPV, germline pathogenic variant; HRD, homologous recombination deficiency; IDFS, invasive disease-free survival; IV, intra-venous; MBC, metastatic breast cancer; mo, months; mPFS, median progression-free survival; ORR, overall response rate; pCR, pathological complete response; Ph, Phase; pts, patients; RCB, residual cancer burden; RCT, randomized clinical trial; RR, response ratio; TNBC, triple-negative breast cancer; TPC, treatment of physician’s choice; mo, months; VCP, carboplatin/paclitaxel + veriparib; VT, veliparib + temozolomide; yr, year.

### 2.2. PALB2

#### 2.2.1. Function 

*PALB2 (partner and localizer of BRCA2)* was originally identified as a gene that produces a BRCA2-interacting protein [98] and was subsequently shown to interact with BRCA1. Consistent with their pivotal roles in maintaining genomic integrity via involvement in HR and DNA repair, mono-allelic *PALB2* GPVs result in an increased risk of BC, whereas bi-allelic *PALB2* GPVs cause FA [99,100,101]. 

PALB2, which is recruited to sites of DNA damage upon phosphorylation in response to DSBs, serves as a molecular scaffold for the formation of the BRCA complex (BRCA1-PALB2-BRCA2-RAD51), thereby facilitating RAD51-mediated strand invasion during HR [75,98,100,102] (Figure 3a).

PALB2 also functions alongside *BRCA1/2* in DNA replication fork protection (Figure 3c).

#### 2.2.2. Prevalence and Risk of Developing Cancers

Approximately 0.5% of patients with BC harbor GPVs in the *PALB2* gene. Although *PALB2* was previously considered a risk gene for BC at moderate penetrance, recent evidence suggests that *PALB2* should be placed in the high-risk category. In two large-cohort studies, the risk of developing BC in *PALB2* GPV carriers was estimated to overlap with that in the *BRCA2* GPV carriers (odds ratios = 5.02 and 3.83 reported by Dorling et al. and Hu et al., respectively) (Table 1). Additionally, the *PALB2* GPV carriers showed a strong association with developing ER-positive BC [12,13]. The absolute lifetime risk of developing BC in the *PALB2* GPV carriers exceeded 30% [12,13], but this might have been dependent on the BC family history. For example, the absolute lifetime risk of developing BC in women at 70 years was as high as 58% for the *PALB2* GPV carriers with two or more first-degree relatives who developed BC under the age of 50 years [103].

Recently, *PALB2* has also been reported to be associated with increased risks of OC and pancreatic cancer [104,105]. The lifetime risk of developing OC is 3–5% for *PALB2* GPV carriers, whereas that for pancreatic cancer is 5–10% [14]; however, the absolute risks are not well estimated and are limited.

#### 2.2.3. Medical Management for Cancer Prevention 

For BC screening in *PALB2* GPV carriers, the NCCN guidelines recommend annual mammograms with consideration of tomosynthesis and breast MRI with contrast from the age of 30. Similarly to that for *BRCA1/2*, the RRM option can be discussed. Evidence of RRSO in the management of OC risk is insufficient, and RRSO may be considered after menopause or earlier if there is a family history of OC [14] (Table 2). Emerging data indicate the efficacy of pancreatic cancer screening in selected individuals at an increased risk of pancreatic cancer.

#### 2.2.4. Treatment

Platinum-based Anticancer Agents

Previous studies have described a responsiveness to platinum agents in pancreatic cancer patients with *PALB2* GPVs and ovarian cancer patients with GPV in HR genes [106,107]. In MBC, a case series of two patients with *PALB2* GPVs who experienced an excellent clinical response to platinum agents has also been reported [108]. One patient with a *PALB2* GPV treated with carboplatin had a complete response and a prolonged duration of response of 30 months. Another patient treated with carboplatin also had a near-complete response for 2 months before discontinuing due to thrombocytopenia. These limited case reports do not provide significant evidence of efficacy, but potentially generate hypotheses of efficacy.

PARP Inhibitors

An expanded phase II study of olaparib for MBC with GPVs in HR-related genes reported that PARP inhibition was sensitive to BC with *PALB2* GPVs [109]. In a phase II study of talazoparib, patients with advanced BC who had GPVs in HR repair pathway genes other than *BRCA1/2* were enrolled, and two of the three responders had GPVs in *PALB2* [110]. The results were consistent with the close interaction between PALB2 and BRCA1/2 in the HR DNA repair pathway. 

Therefore, platinum-based anticancer agents and PARP inhibitors should be considered for the treatment of BC patients with *PALB2* GPVs (Table 2).

## 3. Moderate-Risk Genes for Breast Cancer

In addition to the most well-studied high-penetrance genes, *BRCA1* and *BRCA2*, several other genes that are correlated with a moderately increased risk of developing BC have been identified. These moderate-penetrance genes encoding proteins involved in DSB repair are *CHEK2*, *ATM*, *BARD1*, *RAD51C*, and *RAD51D* [111].

However, their clinical utility for medical management, such as preventive and therapeutic approaches based on a risk assessment for each gene, has not yet been established [14].

### 3.1. ATM

#### 3.1.1. Function

The *ATM* (ataxia-telangiectasia mutated) gene encodes a protein kinase at the peak of a signaling cascade that responds to DSBs and is required to coordinate the resulting cellular responses, such as DDR, cell cycle arrest, and/or apoptosis [112,113]. Once activated by DBS, ATM phosphorylates many downstream effectors, including BRCA1, p53 encoded by *TP53*, and Chk2 encoded by *CHEK2*, which participate in different stages of the DDR [23,24]. ATM involvement in the DDR can explain many pathological hallmarks of ataxia-telangiectasia, including a predisposition to cancer, hypersensitivity to ionizing radiation, immunodeficiency, and infertility. ATM activation upon the formation of DSBs also promotes cell cycle arrest through the activation of cell cycle checkpoints at multiple stages of the cell cycle, including G1, S, and G2 [114] (Figure 3a,b). Alternatively, ATM can induce apoptosis through p53 and Chk2, depending on the cell type and level of genomic damage.

#### 3.1.2. Prevalence and Risk of Developing Cancers

In two recent large studies, the prevalence of GPVs in *ATM* reported by Dorling et al. and Hu et al. was 0.60% and 0.78%, respectively, in unselected BC patients, whereas that in unaffected women was 0.29% and 0.41%, respectively [12,13] (Table 1). This indicates that heterozygote *ATM* GPV carriers have an approximately two-fold higher risk of developing BC than non-carriers (odds ratios = 2.10 and 1.82, respectively) [12,13], with an absolute BC lifetime risk of 15–40% [12,13,115,116]. For protein-truncating variants in *ATM*, the odds ratio of ER-positive disease is higher than that of ER-negative disease [12]. Some evidence has indicated that certain germline missense variants in *ATM* may act in a dominant-negative manner to increase the risk of BC compared with truncating variants. Notably, carriers of c.7271T > G (p.Val2424Gly) are reportedly associated with a higher risk of invasive ductal BC (odds ratio = 3.76) [117] and have a 69% risk of developing BC by 70 years [116].

*ATM* GPV carriers are also associated with an increased risk of pancreatic cancer (odds ratio = 4.21), prostate cancer (odds ratio = 2.58), gastric cancer (odds ratio = 2.97), ovarian cancer (odds ratio = 1.57), colorectal cancer (odds ratio = 1.49), and melanoma (odds ratio = 1.46) [117].

Homozygous or compound heterozygous *ATM* GPVs cause a rare autosomal recessive disorder called A-T, characterized by progressive cerebellar ataxia, telangiectasia, oculomotor apraxia, immunodeficiency, cancer susceptibility, and radiation sensitivity [118]; therefore, counseling *ATM* GPV carriers against the risk of autosomal recessive conditions in their offspring is recommended [14].

#### 3.1.3. Medical Management for Cancer Prevention

For *ATM* GPV carriers, annual mammograms with consideration of tomosynthesis beginning at 40 years and consideration of breast MRI with contrast are recommended by the NCCN guidelines. As there is insufficient evidence to recommend RRM and RRSO, the procedures are considered based on family history (Table 2). Emerging data indicate the efficacy of pancreatic cancer screening in selected individuals at an increased risk of pancreatic cancer. In the absence of family history, screening for cancers other than BC is not recommended [14].

#### 3.1.4. Treatment

Platinum-based Anticancer Agents

In a case series on pancreatic ductal adenocarcinoma, 8 out of 10 pancreatic cancer patients with *ATM*, *ATR,* or *CHEK2* GPVs were treated with an oxaliplatin-based chemotherapy regimen and 5 patients demonstrated a partial response or stable disease [119]. The clinical experience of platinum-based agents in patients with *ATM* GPVs is very limited.

CDK4/6 Inhibitors

Recent studies have shown that the ATM-Chk2-Cdc25A pathway participates in resistance to CDK4/6 inhibitors [120,121,122]. ATM activates Chk2 as a DNA damage sensor, and Chk2 phosphorylates and degrades Cdc25A, a phosphatase that may inhibit the phosphorylation of CDK4/6 [120,121]; therefore, the activation of Cdc25A through a deficiency of ATM-Chk2 signaling may reactivate the CDK4/6 complex. Retrospective data suggested that 15 advanced BC patients with GPVs in HR genes (*BRCA1/2*-*ATM*-*CHEK2*) treated with CDK4/6 inhibitors had poor outcomes [83]; however, this is limited data to suggest CDK4/6 inhibitor resistance in patients with GPVs in the HR genes. CDK4/6 inhibitors plus endocrine therapy would still be recommended as the first-line treatment for ER-positive/HER 2-negative MBC with *ATM* GPVs.

PARP Inhibitors

Considering the efficacy of platinum-based agents and PARP inhibitors in treating *BRCA1/2*-associated tumors, a similar efficacy can be expected in BC patients with GPVs in *ATM*. Nevertheless, in an expanded phase II study of olaparib for MBC with GPVs in HR-related genes, no response was observed in patients with GPVs in either *ATM* or *CHEK2* alone, although the reliability of the results is limited owing to the small sample size [109].

In a phase II clinical trial, olaparib combined with paclitaxel has shown a significant improvement in OS in patients with advanced gastric cancer, especially in those with *ATM*-deficient tumors [123]; however, the subsequent phase III trial with 525 advanced gastric cancer patients did not show a significant improvement in OS with olaparib in the overall or patients with *ATM*-deficient tumors (12.0 months vs. 10.0 months, hazard ratio = 0.73 and *p* = 0.25 in the ATM-negative population) [124].

The PROfound trial has shown that olaparib led to significantly longer PFS and OS than the physician’s choice of enzalutamide or abiraterone among men with metastatic castration-resistant prostate cancer who had at least one germline or somatic PV in *BRCA1*, *BRCA2*, or *ATM* in tumors. 

However, exploratory analyses suggest that patients with *BRCA1/2* GPVs derived the most benefit, and showed the hazard ratios for PFS and OS among patients with *ATM* GPVs (olaparib vs. control) were 1.04 (95% CI, 0.61 to 1.87) and 0.93 (95% CI, 0.53 to 1.75), respectively [125,126].

Radiation Therapy

The high sensitivity of A-T to radiation raises numerous concerns regarding the use of breast radiation therapy (RT) in *ATM* GPV heterozygous carriers.

A study of 135 patients with BC, including 20 *ATM* GPV carriers treated with RT after breast-conserving surgery (BCS), showed no significant differences in local recurrence or metastasis-free survival between the carriers and non-carriers [127]. McDuff et al. reviewed the association of *ATM* variants with radiation-induced toxicity or the risk of secondary malignancy [128] and concluded that adjuvant RT is safe for most BC patients with GPVs in *ATM*. The possible exceptions are patients with the c.5557G > A variant, in whom a small but increased risk for the development of both acute and late radiation effects has been identified. Furthermore, patients younger than 45 years with certain rare deleterious *ATM* variants may have an increased risk of developing CBC. Heterozygous *ATM* GPV should be recommended to not avoid RT [14,128].

Mammogram screening benefits and harms for young *ATM* GPV carriers are ongoing and remain controversial [50,129]. Mammogram in addition to MRI screening in young *ATM* GPV carriers is of little benefit because mammogram rarely detects BC missed on MRI [130], and could be harmful due to the possible risk of secondary malignancy [129]. A recent study suggested that mammograms could be delayed until age 40 years for *ATM* GPV carriers when screening with MRI [129].

### 3.2. CHEK2

#### 3.2.1. Function

*CHEK2* is a tumor suppressor gene that encodes protein Chk2 (checkpoint kinase 2), a serine/threonine kinase [131]. Chk2 is activated by ATM in response to DSB or replicative stress and then phosphorylates numerous downstream substrates, including p53 encoded by *TP53* and BRCA1, which are involved in various cellular pathways that activate checkpoint-mediated cell cycle arrest, DNA repair, and apoptosis [132] (Figure 3b); thus, Chk2 may function as a tumor suppressor by slowing the cell cycle progression to allow sufficient time for DNA repair or by inducing cell death to eliminate genomically unstable cells.

Several putative GPVs have been reported in *CHEK2* [133]. Most studies have focused on the frameshift variant NM_007194.4:c.1100delC (p.T367fs) and missense variant c.470T > C (p.I157T), which correlate with a moderate and low BC risk, respectively [132,134].

#### 3.2.2. Prevalence and Risk of Developing Cancers

In two recent large studies, the prevalence of *CHEK2* GPVs was reported to be 1.44% and 1.08% in BC patients and 0.62% and 0.42% in unaffected control women by Dorling et al. and Hu et al., respectively [10,11]. *CHEK2* GPVs were significantly associated with BC (odds ratio = 2.54 and 2.47, respectively). The *CHEK2*:c.1100delC variant showed a higher relative risk (odds ratio = 2.66) than the other truncating and missense GPVs [12,13]. The cumulative lifetime BC risk of certain variants, namely, c.1100delC and p.I157T, ranges from 28% to 37%, depending on the family history [15,135]. For protein-truncating variants in *CHEK2*, the odds ratio for ER-positive disease is higher than that for ER-negative disease [12].

*CHEK2* GPVs, particularly c.1100delC and p.I157T, are also associated with increased risks of other types of cancer, such as colorectal, prostate, kidney, and thyroid cancer [15,135,136].

#### 3.2.3. Medical Management for Cancer Prevention 

As the risk data are based solely on frameshift GPVs in *CHEK2*, the management should be based on the best estimates of the carcinogenic risk for a particular GPV. For *CHEK2* GPV carriers, an annual mammogram with consideration of tomosynthesis starting at 40 years and breast MRI with contrast is recommended by the NCCN guidelines. Since sufficient data are not available on the benefits of RRM, prophylactic surgery is considered based on family history (Table 2). 

For *CHEK2* GPV carriers unaffected by colorectal cancer, high-quality colonoscopy is proposed every 5 years starting at 40 years or 10 years prior to the first-degree relative’s age with colorectal cancer if the carriers have a first-degree relative with colorectal cancer. For *CHEK2* GPV carriers, there is no specific screening strategy for cancers other than breast and colorectal cancer [14]. 

#### 3.2.4. Treatment

Anthracycline-based Chemotherapy/Tamoxifen

The LOF of the ATM-Chk2-p53 cascade due to GPVs in *CHEK2* or *TP53* has been reported to be associated with a resistance to anthracycline-based chemotherapy in BC patients [137]. In contrast, no difference in the effects of chemotherapy and endocrine therapy on MBC has been observed between germline *CHEK2* 1100delC and germline *CHEK2* wild-type carriers [138].

Platinum-based Anticancer Agents

A limited case series have described that five out of eight pancreatic cancer patients with GPVs in *ATM*, *ATR*, or *CHEK2* demonstrated a clinical benefit to oxaliplatin-based chemotherapy [119]; however, to our knowledge, there are no previous reports demonstrating efficacy to platinum agents in BC patients with *ATM* GPVs.

PARP Inhibitors

In an expanded phase II study of olaparib for MBC with GPVs in HR-related genes, no response was observed in BC patients with *ATM* or *CHEK2* GPVs alone, although the robustness of the results was limited due to the small sample size [109].

In exploratory analyses of the PROfound trial evaluating the efficacy of olaparib compared with the treatment of physician’s choice in patients with metastatic castration-resistant prostate cancer who had disease progression while receiving a new hormonal agent, the hazard ratio for PFS among the patients with *CHEK2* GPVs was 0.87 (95% CI, 0.23 to 4.13) [125].

The sensitivity to specific treatment regimens in BC patients with *CHEK2* GPVs is unclear, and the utility of *CHEK2* as a companion test for predicting treatment sensitivity has not been established.

### 3.3. BARD1

#### 3.3.1. Function

*BARD1* (BRCA1 associated RING domain 1) encodes a ubiquitin ligase that interacts with the N-terminal region of BRCA1. BRCA1-BARD1 heterodimerization is required for their mutual stability, HR function, and tumor suppressor activity [16,139,140] (Figure 3a).

#### 3.3.2. Prevalence and Risk of Developing Cancers

In a recent population-based study, protein-truncating GPVs in *BARD1* were detected in 0.13% and 0.06% of BC patients and controls, respectively (odds ratio = 2.09, *p* = 0.00098) [12] (Table 1). The GPVs in *BARD1* were associated with a moderate risk of ER-negative BC and TNBC, but not ER-positive BC [13].

In addition, GPVs in *BARD1* have been reported to be associated with an increased risk of OC, although studies estimating the familial risk of *BARD1* GPVs in OC are still lacking [13,141].

#### 3.3.3. Medical Management for Cancer Prevention 

Similar to those for patients carrying other moderate-risk genes, such as *CHEK2* and *ATM*, annual mammograms with consideration of tomosynthesis beginning at 40 years, along with the consideration of breast MRI with contrast, are recommended for *BARD1* GPV carriers. RRM is not recommended for *BARD1* GPV carriers because of insufficient evidence; however, the procedure should be considered based on the family history [14].

### 3.4. RAD51C/RAD51D

#### 3.4.1. Function

RAD51 is a key protein that mediates HR and forms a complex with a family of accessory proteins, known as RAD51 paralogs. *RAD51C* and *RAD51D* encode RAD51 paralogs that interact with the BRCA1 and BRCA2 proteins and support the DNA repair process, particularly the HR repair pathway. BRCA2 has eight RAD51-binding domains called BRC repeats, which can carry multiple RAD51s to DNA injury sites, leading to strand invasion and HR by RAD51 [142,143] (Figure 3a).

GPVs in *RAD51C* and *RAD51D* predispose carriers to BC and OC [10,144].

#### 3.4.2. Prevalence and Risk of Developing Cancer

In a recent population-based study, GPVs in *RAD51C* and *RAD51D* were detected in 0.11% and 0.10% of BC cases and 0.05% and 0.04% of controls, respectively (odds ratio = 1.93 and 1.80, respectively; *p* = 0.0070 and 0.018, respectively) [12] (Table 1). 

The estimated lifetime absolute risks of BC for *RAD51C* and *RAD51D* GPV carriers were 15–40% [12,13,14]. Notably, GPVs in *RAD51C* and *RAD51D* had a stronger association with TNBC (odds ratio = 5.71 and 6.01, respectively), than with ER-positive BC (odds ratio = 1.31 and 1.52, respectively) [12].

GPVs in *RAD51C* and *RAD51D* are also associated with an increased risk of developing OC (odds ratio = 5.2 and 12, respectively, and *p* = 0.035 and 0.019, respectively) [145]. The lifetime risk of OC has been estimated to be >10% for *RAD51C* and *RAD51D* GPV carriers [14,145,146].

#### 3.4.3. Medical Management for Cancer Prevention 

Owing to insufficient evidence, BC screening is based on family history. In contrast, the NCCN guidelines recommend that RRSO should be considered beginning at 45–50 years of age or earlier, based on a specific family history of an earlier onset of OC [14] (Table 2).

## 4. Conclusions

This review has summarized the molecular functions of key BC-associated CPGs, namely, *BRCA1*, *BRCA2*, *PALB2*, *ATM*, *BARD1*, *CHEK2*, *RAD51C*, and *RAD51D*, as well as the clinical management of individuals harboring GPVs in these genes. 

The penetrance of GPVs in each gene varies, reflecting the distinct risks of BC susceptibility. Although GPVs in *BRCA1/2* convey the highest genetic risk of BC and data on the prevention and treatment of BCs associated with GPVs in *BRCA1/2* are accumulating, optimal clinical strategies for BCs associated with GPVs in CPGs other than *BRCA1/2* have not yet been established. Most BC-associated CPGs participate in the DNA damage repair pathways and cell cycle checkpoint mechanisms and work together in such cascades; therefore, a fundamental understanding of the pathogenic cascade caused by GPVs in BC-associated CPGs may enable us to accurately estimate the genetic risk of developing BC and to select the appropriate preventive and therapeutic strategies for treating hereditary BCs. 

In addition, a refinement of the risk models using polygenic risk scores may enable a better definition of the personalized risks for hereditary BCs [147], with an enhanced quality of clinical management offered.

For rare genes where there is limited data for targeted therapeutics in each trial, sharing data across trials is the only way to enable an effective analysis of the therapeutic response, and is key to future trial development.

## Figures and Tables

**Figure 1 ijms-23-07481-f001:**
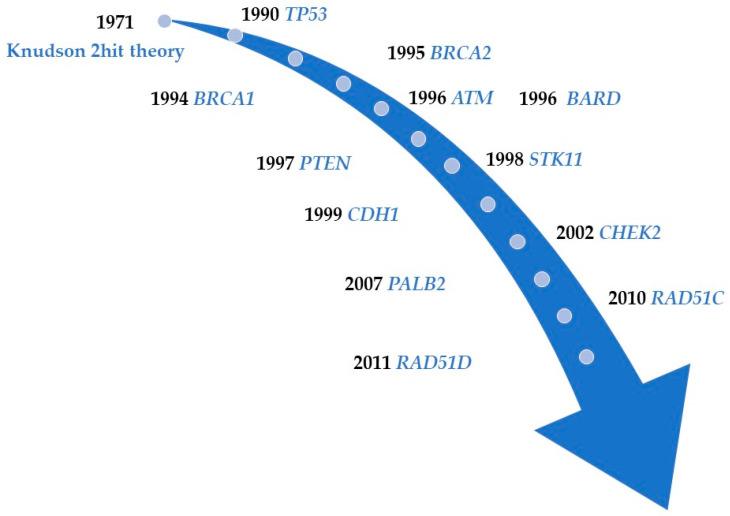
Timeline reflecting the identification of 12 breast cancer predisposition genes.

**Figure 2 ijms-23-07481-f002:**
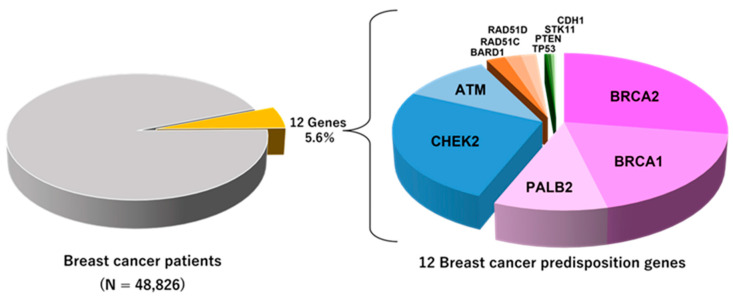
Frequency of protein-truncating variants in 12 breast cancer predisposition genes in the population-based study reported by Dorling et al. [12].

**Figure 3 ijms-23-07481-f003:**
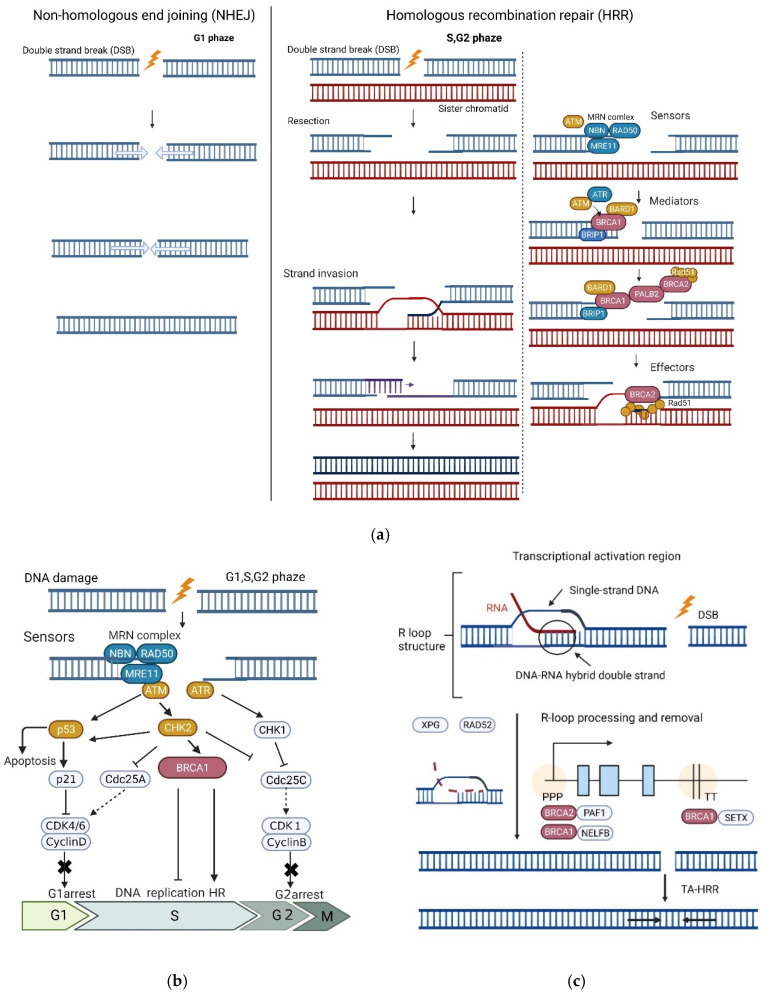
Distinct functions of BRCA1 and BRCA2 in (**a**) homologous recombination (HR) repair, (**b**) DNA damage cell cycle checkpoint, (**c**) R-loop processing and transcription, and (**d**) DNA replication fork protection.

**Figure 4 ijms-23-07481-f004:**
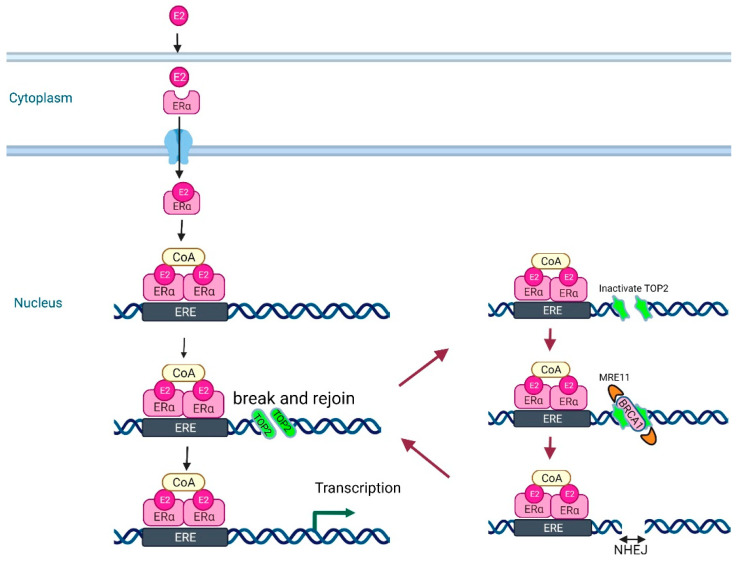
Binding of estrogens (E2s) to estrogen receptor (ERα) transiently induces DNA double-strand breaks via topoisomerase II (TOP2). BRCA1 ensures genome integrity by removing the pathological TOP2–DNA complexes induced by estrogen. Abbreviations: ERE, estrogen responsive element; CoA, co-activator; NHEJ, non-homologous end-joining.

**Figure 5 ijms-23-07481-f005:**
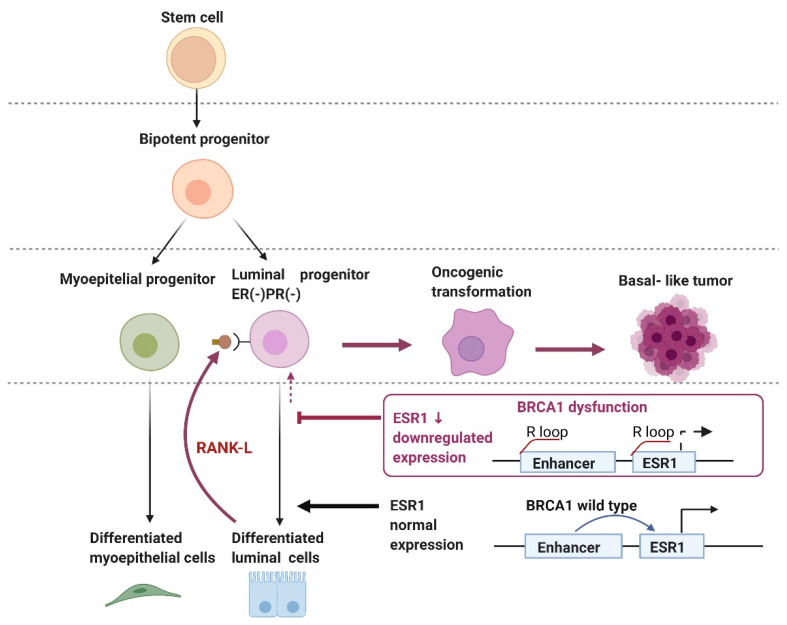
Carcinogenic mechanism of basal-like breast cancer (BC). Dysregulation of ESR1 transcription due to BRCA1 dysfunction inhibits luminal progenitor cell differentiation. RANK/RANK-L-mediated proliferation of BRCA1-deficient cells leads to the generation of driver mutations, resulting in the development of basal-like BC.

**Table 1 ijms-23-07481-t001:** Prevalence of GPVs of 12 BC predisposition genes in BC patients and control subjects.

Susceptibility	Gene	BC Lifetime Risk ^1^	Prevalence of GPVs		Odds Ratio (95% CI)			
			Women with BC	Controls	All BC	ER-Positive BC	ER-Negative BC	Triple-Negative BC
High	*BRCA1*	>60%	1.05% ^2^	0.11% ^2^	10.57 (8.02–13.93) ^2^	3.92 (2.82–5.43) ^2^	35.32 (26.21–47.60) ^2^	56.80 (41.18–78.34) ^2^
			0.85% ^3^	0.11% ^3^	7.62 (5.33–11.27) ^3^	3.39 (2.17–5.45) ^3^	26.33 (17.28–41.52) ^3^	42.88 (26.56–71.25) ^3^
	*BRCA2*	>60%	1.54% ^2^	0.26% ^2^	5.85 (4.85–7.06) ^2^	5.69 (4.65–6.96) ^2^	7.53 (5.89–9.62) ^2^	11.19 (8.27–15.16) ^2^
			1.29% ^3^	0.24% ^3^	5.23 (4.09–6.77) ^3^	4.66 (3.52–6.23) ^3^	8.89 (6.36–12.47) ^3^	9.70 (5.97–15.47) ^3^
	*PALB2*	41–60%	0.56% ^2^	0.10% ^2^	5.02 (3.73–6.76) ^2^	4.45 (3.23–6.14) ^2^	6.72 (4.54–9.95) ^2^	10.36 (6.42–16.71) ^2^
			0.46% ^3^	0.12% ^3^	3.83 (2.68–5.63) ^3^	3.13 (2.02–4.96) ^3^	9.22 (5.63–15.25) ^3^	13.03 (7.08–23.75) ^3^
Moderate	*CHEK2*	15–40%	1.44% ^1^	0.62% ^2^	2.54 (2.21–2.91) ^2^	2.67 (2.30–3.11) ^2^	1.64 (1.25–2.16) ^2^	1.06 (0.63–1.76) ^2^
			1.08% ^3^	0.42% ^3^	2.47 (2.02–3.05) ^3^	2.60 (2.05–3.31) ^3^	1.40 (0.83–2.25) ^3^	1.63 (0.72–3.20) ^3^
	*ATM*	15–40%	0.60% ^2^	0.29% ^2^	2.10 (1.71–2.57) ^2^	2.33 (1.87–2.91) ^2^	1.01 (0.64–1.59) ^2^	0.91 (0.42–1.95) ^2^
			0.78% ^3^	0.41% ^3^	1.82 (1.46–2.27) ^3^	1.96 (1.52–2.53) ^3^	1.04 (0.59–1.72) ^3^	0.50 (0.12–1.36) ^3^
	*BARD1*	15–40%	0.12% ^2^	0.06% ^2^	2.09 (1.35–3.23) ^2^	1.40 (0.81–2.42) ^2^	5.99 (3.51–10.21) ^2^	9.29 (4.58–18.85) ^2^
			0.15% ^3^	0.11% ^3^	1.37 (0.87–2.16) ^3^	0.91 (0.49–1.64) ^3^	2.52 (1.18–5.00) ^3^	3.18 (1.16–7.42) ^3^
	*RAD51C*	15–40%	0.11% ^2^	0.05% ^2^	1.93 (1.20–3.11) ^2^	1.31 (0.74–2.30) ^2^	3.99 (2.20–7.26) ^2^	5.71 (2.69–12.13) ^2^
			0.13% ^3^	0.11% ^3^	1.20 (0.75–1.93) ^3^	0.83 (0.44–1.54) ^3^	2.19 (0.97–4.49) ^3^	NA^3^
	*RAD51D*	15–40%	0.10% ^2^	0.04% ^2^	1.80 (1.11–2.93) ^2^	1.52 (0.87–2.65) ^2^	2.92 (1.47–5.78) ^2^	6.01 (2.73–13.24) ^2^
			0.08% ^3^	0.04% ^3^	1.72 (0.88–3.51) ^3^	1.61 (0.71–3.70) ^3^	3.93 (1.40–10.29) ^3^	NA ^3^
Syndrome	*TP53*	>60%	0.01% ^2^	0.003% ^2^	3.06 (0.63–14.91) ^2^	1.95 (0.32–11.82) ^2^	5.42 (0.75–39.24) ^2^	NA ^2^
			0.06% ^3^	0.01% ^3^	NA ^3^	NA ^3^	NA ^3^	NA ^3^
	*PTEN*	40–60%	0.02% ^2^	0.01% ^2^	2.25 (0.85–6.00) ^2^	2.42 (0.84–6.97) ^2^	NA ^2^	NA ^2^
			0.02% ^3^	0.01% ^3^	NA^3^	NA^3^	NA ^3^	NA ^3^
	*STK11*	40–60%	0.01% ^2^	0.009% ^2^	1.60 (0.48–5.28) ^2^	1.56 (0.35–7.03) ^2^	NA ^2^	NA ^2^
	*CDH1*	41–60%	0.02% ^2^	0.02% ^2^	0.86 (0.37–1.98) ^2^	1.05 (0.42–2.63) ^2^	1.11 (0.24–5.10) ^2^	1.44 (0.18–11.28) ^2^
			0.05% ^3^	0.02% ^3^	2.50 (1.01–7.07) ^3^	3.37 (1.24–10.72) ^3^	NA ^3^	NA ^3^

^1^ Reference [14] and the National Comprehensive Cancer Network (NCCN) Clinical Practice Guidelines for Genetic/Familial High-Risk Assessment: Breast, Ovarian, and Pancreatic, Version 2.2022 (https://www.nccn.org/home, Last accessed on 16 April 2022). ^2^ Reference [12]: ^3^ Reference [13]. Red numbers indicate significantly increased risk (*p* < 0.05). Abbreviations: BC, breast cancer; CI, confidence interval; ER, estrogen receptor; GPV, germline pathogenic variants; NA, not applicable due to too few events to calculate a stable odds ratio.

**Table 2 ijms-23-07481-t002:** Medical management for cancer prevention as recommended by the NCCN guidelines ^1^.

Susceptibility	Gene	Risk-Reducing Surgery		BC Screening	BC Treatment	Other Cancer Risks
		RRM	RRSO			
High	*BRCA1*	Discuss option of RRM	Recommend RRSO	Age 25 y: annual breast MRI Age 30–75 y: additional mammogram	Platinum agents and PARP inhibitors	Ovary, pancreas, and prostate
	*BRCA2*					Ovary, pancreas, prostate, and melanoma
	*PALB2*		Evidence insufficient, manage based on family history	Age 30 y: annual mammogram and breast MRI		Ovary and pancreas
Moderate	*ATM*	Evidence insufficient for RRM; manage based on family history		Age 40 y: annual mammogram and consider breast MRI	Heterozygous *ATM* GPV should not lead to a recommendation to avoid RT	Ovary and pancreas
	*CHEK2*				(Insufficient evidence)	Colon
	*BARD1*					(Insufficient evidence)
	*RAD51C*	Insufficient data; manage based on family history	Consider RRSO	Insufficient data; managed based on family history		Ovary
	*RAD51D*					
Syndrome	*TP53*	Discuss option of RRM		Age 20 y: annual breast MRI Age 30–75 y: additional mammogram	Therapeutic RT for cancer should be avoided when possible; diagnostic radiation should be minimized to the extent feasible without sacrificing accuracy	Adrenocortical gland, central nervous system, bone, and soft tissue
	*PTEN*			Age 30–75 y: annual mammogram and breast MRI	(Insufficient evidence)	Thyroid, kidney, endometrium, and colon
	*CDH1*			Age 30 y: annual mammogram and consider breast MRI		Stomach
	*STK11*	Evidence insufficient for RRM; manage based on family history	No established data	Age 30 y: annual mammogram and breast MRI		Colon, stomach, small bowel, pancreas, cervix, uterus, ovary, testis, and lung

^1^ Reference [14] and the National Comprehensive Cancer Network (NCCN) Clinical Practice Guidelines for Genetic/Familial High-Risk Assessment: Breast, Ovarian, and Pancreatic, Version 2.2022 (https://www.nccn.org/home, Last accessed on 16 April 2022). Gray background means no established association between GPVs in each gene and ovarian cancer. Abbreviations: BC, breast cancer; MRI, magnetic resonance imaging; PARP, poly (ADP-ribose) polymerase; RRM, risk-reducing mastectomy; RRSO, risk-reducing salpingo-oophorectomy; RT, radiotherapy.

## Data Availability

Not applicable.

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
