# Peer review of "Functions of Breast Cancer Predisposition Genes: Implications for Clinical Management"

_ijms, 2022, doi:10.3390/ijms23137481_

Round 1
Reviewer 1 Report
Please find my comments attached.

Reviewer 2 Report
This article entitled "Functions of Breast Cancer Predisposition Genes: Implications for Clinical Management" describes the molecular functions of breast cancer predisposition genes in detail. Indeed, the explanation of molecular mechanism, including schemas, are valuable and persuasive. However, some points are inadequate to acceptance. The authors should consider targeted genes again, and should improve the following points at least;
1) In line 47, the authors describe that “Dorling et al. included 34 putative BC-associated CPGs”. However in Figure2., only 12 Breast cancer predisposition genes are selected. The authors are encouraged to describe this discrepancy.
2) In Table1., “Red numbers indicate significantly increased risk (P < 0.05)” is described in the footnote, however, no red number is seen. The authors should check the description.
3) In line 79, authors describe that some hereditary tumors “were excluded from this review”. The authors are recommended to explain why those syndromes were excluded. It is thought that the reader's understanding can be obtained by having a clear explanation, such as mentioning only HRD related genes.
4) In Table2., STK11 and CDH1 are easier to understand if the lines are swapped. It should be described about what gray background means in the table. According to NCCN guidelines’ recommendation, BRIP1 and NF1 are also candidate genes to be described.
Especially BRIP1, authors are encouraged to add description in “3. Moderate-Risk Genes for Breast Cancer” section.
5). In line 139, it is described that “aldehyde accumulation may modulate tissue-specific cancer progression in BRCA1/2 GPV carriers”. According to the above description, only BRCA2 seems to be related to aldehyde, so please check if it is correct.
6). In Figure3., FA complex is mentioned in the article, but not in the figure. The authors are encouraged to add FA complex in Figure3.
7). In Figure4., the following abbreviations should be included in the figure legend; ERE, CoA, NHEJ.
8). In Figure5., The Stem cell shows BRCA-/-, does this mean the BRCA1/2 wild type? Calling BRCA to refer to both BRCA1/2 is not defined in the article, and is considered inappropriate. The authors should change BRCA-/- to more suitable description.
9). In line 298, yellow mark remains. It should be deleted.
Reviewer 3 Report
I have reviewed the article “Functions of Breast Cancer Predisposition Genes: Implications for Clinical Management” submitted to the International Journal of Molecular Sciences. The
review has already been written nicely. It provides evidence of the functions of key BC-associated CPGs and strategies for the
clinical management of individuals harbouring GPVs of such genes.
Based on the quality of the article, I recommend publishing it after addressing a minor revision.
1) Abstract and introduction are good. In the introduction, it would be interesting to indicate the implication of the BRCA methylation state (DOI: 10.3390/cancers13061391)
2)The drafting of the text is well done and schematized. Punctuation should be revised.
3)The current form of Conclusions is a little simple. It would be better to expand it with more essential details.
4)Please double check the whole manuscript for potential grammar errors and typos.
5)The English writing needs further proofreading. There are still some linguistic errors and vague descriptions remaining in the manuscript.
6) Image quality and definition should be improved
Round 2
Reviewer 1 Report
This reviewed version appears to meet many but all of the suggested changes/ edits. It is an extensive review of the current literature.
For genes where there is limited data for targeted therapies, I would suggest that they could supplement their paper with references to whether responses were seen in other tumour types to the same targeted agent (Parp inhibitor, platinum chemotherapy) with the same pathogenic variant- ATM, CHEK2 etc. This will add weight and validity to the conclusions.
Where limited data is published- the authors should acknowledge this. Case reports are potentially hypothesis generating, but by no means represent significant evidence for efficacy.
Conclusions for the rare genes could be a little bolder- the importance of being able to share data across trials is the only way to enable effective analysis for response. In future trial development, it is imperative that this is built into the capacity for trials; pharma should be collaborative and accountable for enabling the improvement of outcomes for rare cancers where individual trials are not feasible.
